# MS MARCO: A HUMAN GENERATED MACHINE READING COMPREHENSION DATASET

**Tri Nguyen, Mir Rosenberg, Xia Song, Jianfeng Gao, Saurabh Tiwary,**
**Rangan Majumder and Li Deng**
Microsoft AI & Research
Bellevue, WA, USA
{trnguye,miriamr,xiaso,jfgao,satiwary,ranganm,deng}@microsoft.com

## ABSTRACT

This paper presents our work on the design and development of a new, large scale dataset, which we name MS MARCO, for MAchine Reading COmprehension. This new dataset is aimed to overcome a number of well-known weaknesses of previous publicly available datasets for the same task of reading comprehension and question answering. In MS MARCO, all questions are sampled from real anonymized user queries. The context passages, from which answers in the dataset are derived, are extracted from real web documents using the most advanced version of the Bing search engine. The answers to the queries are human generated. Finally, a subset of these queries has multiple answers. We aim to release one million queries and the corresponding answers in the dataset, which, to the best of our knowledge, is the most comprehensive real-world dataset of its kind in both quantity and quality. We are currently releasing 100,000 queries with their corresponding answers to inspire work in reading comprehension and question answering along with gathering feedback from the research community.

## 1 INTRODUCTION

Building intelligent agents with the ability for reading comprehension (RC) or open-domain question answering (QA) over real world data is a major goal of artificial intelligence. Such agents can have tremendous value for consumers because they can power personal assistants such as Cortana web (b), Siri web (e), Alexa web (a), or Google Assistant web (d) found on phones or headless devices like Amazon Echo web (c), all of which have been facilitated by recent advances in deep speech recognition technology Hinton et al. (2012); Dahl et al. (2012). As these types of assistants rise in popularity, consumers are finding it more convenient to ask a question and quickly get an answer through voice assistance as opposed to navigating through a search engine result page and web browser. Intelligent agents with RC and QA abilities can also have incredible business value by powering bots that automate customer service agents for business found through messaging or chat interfaces.

Real world RC and QA is an extremely challenging undertaking involving the amalgamation of multiple difficult tasks such as reading, processing, comprehending, inferencing/reasoning, and finally summarizing the answer.

The public availability of large datasets has led to many breakthroughs in AI research. One of the best examples is ImageNet's Deng et al. (2009) exceptional release of 1.5 million labeled examples and 1000 object categories which has led to better than human level performance on object classification from images He et al. (2015). Another example is the very large speech databases collected over 20 years by DARPA that enabled successes of deep learning in speech recognition Deng & Huang (2004). Recently there has been an influx of datasets for RC and QA as well. These databases, however, all have notable drawbacks. For example, some are not large enough to train deep models Richardson et al. (2013), and others are larger but are synthetic.

One characteristic in most, if not all, of the existing databases for RC and QA research is that the distribution of questions asked in the databases are not from real users. In the creation of most RC or QA datasets, usually crowd workers are asked to create questions for a given piece of text or

document. We have found that the distribution of actual questions users ask intelligent agents can be very different from those conceived from crowdsourcing them from the text.

Furthermore, real-world questions can be messy: they may include typos and abbreviations. Another characteristic of current datasets is that text is often from high-quality stories or content such as Wikipedia. Again, real-world text may have noisy or even conflicting content across multiple documents and our experience is that intelligent agents will often need to operate over this type of problematic data.

Finally, another unrealistic characteristic of current datasets is that answers are often restricted to an entity or a span from the existing reading text. What makes QA difficult in the real world is that an existing entity or a span of text may not be sufficient to answer the question. Finding the best answer as the output of QA systems may require reasoning across multiple pieces of text/passages. Users also prefer answers that can be read in a stand-alone fashion; this sometimes means stitching together information from multiple passages, as the ideal output not only answers the question, but also has supporting information or an explanation.

In this paper we introduce Microsoft MAchine Reading COmprehension (MS MARCO) - a large scale real-world reading comprehension dataset that addresses the shortcomings of the existing datasets for RC and QA discussed above. The questions in the dataset are real anonymized queries issued through Bing or Cortana and the documents are related web pages which may or may not be enough to answer the question. For every question in the dataset, we have asked a crowdsourced worker to answer it, if they can, and to mark relevant passages which provide supporting information for the answer. The answer is strongly encouraged to be in the form of a complete sentence, so the workers may write a longform passage on their own. MS MARCO includes 100,000 questions, 1 million passages, and links to over 200,000 documents. Compared to previous publicly available datasets, this dataset is unique in the sense that (a) all questions are real user queries, (b) the context passages, which answers are derived from, are extracted from real web documents, (c) all the answers to the queries are human generated, (d) a subset of these queries has multiple answers, (e) all queries are tagged with segment information.

## 2  RELATED WORK

| Dataset | Segment | Query Source | # Queries | # Documents |
|---------|---------|--------------|-----------|-------------|
| MCTest | N | crowdsourced | 2640 | 660 |
| WikiQA | N | User logs | 3047 | 29,258 sentences |
| CNN/Daily Mail | N | Cloze | 1.4M | 93K CNN, 220K Daily Mail |
| Children's Book | N | Cloze | 688K | 688K contexts from 108 books |
| SQuAD | N | Crowdsourced | 100K | 536 |
| MS MARCO | Y | User logs | 100K | 1M passages, 200K+ documents |

Table 1: Comparison of some properties of existing datasets vs MS MARCO. MS MARCO is the only large dataet with open ended answers from real user queries

Datasets have played a significant role in making forward progress in difficult domains. The ImageNet dataset Deng et al. (2009) is one of the best known for enabling advances in image classification and detection and inspired new classes of deep learning algorithms Krizhevsky et al. (2012) Girshick et al. (2014) He et al. (2015). Reading comprehension and open domain question answering is one of those domains existing systems still struggle to solve Weston et al. (2015). Here we summarize a couple of the previous approaches towards datasets for reading comprehension and open domain question answering.

One can find a reasonable amount of semi-synthetic reading comprehension and question answering datasets. Since these can be automatically generated they can be large enough to apply modern data intensive models. Hermann et al. created a corpus of cloze style questions from CNN / Daily News summaries Hermann et al. (2015) and Hill et al. has built the Children's Book Test Hill et al. (2015). Another popular question answering dataset involving reasoning is by Weston et al. Weston et al. (2015). One drawback with these sets is it does not capture the same question characteristics we find with questions people ask in the real world.

MCTest is a challenging dataset which contains 660 stories created by crowdworkers, 4 questions per story, and 4 answer choices per question Richardson et al. (2013), but real-world QA systems needs to go beyond multiple choice answers or selecting from known responses. WikiQA is another set which includes 3047 questions Yang et al. (2015). While other sets are synthetic or editor-generated questions WikiQA is constructed using a more natural process using actual query logs. It also includes questions for which there are no correct sentences which is an important component in any QA system. Unfortunately, these sets are too small to try data demanding approaches like deep learning.

A more recently introduced reading comprehension dataset is the Stanford Question Answering Dataset (SQuAD) Rajpurkar et al. (2016) which consists of 107785 question/answer pairs from 536 articles where the answer is span of paragraph. A few differences between MS MARCO and SQuAD is (a) SQuAD consisting of questions posed by crowdworkers while MS MARCO is sampled from the real world, (b) SQuAD is on a small set of high quality Wikipedia articles while MS MARCO is from a large set of real web documents, and (c) SQuAD consists of spans while MS MARCO has human generated answers (if there is one).

## 3 THE MS MARCO DATASET

In order to deliver true machine Reading Comprehension (RC), we start with QA as the initial problem to solve. Our introduction covered some of the key advantages of making very large RC or QA datasets freely available that contain only real-world questions and human crowdsourced answers versus artificially generated data. Given those advantages, our goal is that MS MARCO - a large scale, real-world and human sourced QA dataset - will become a key vehicle to empower researchers to deliver many more AI breakthroughs in the future, just like ImageNet Deng et al. (2009) enabled for image comprehension before.

Additionally, building an RC-oriented dataset helps us understand a contained yet complex RC problem while learning about all of the infrastructure pieces needed to build such a large one-million query set that helps the community make progress on state-of-the-art research problems. This task is also helping us experiment with natural language processing and deep learning models as well as to understand detailed characteristics of the very large training data required to deliver a true AI breakthrough in RC.

This first MS MARCO release contains 100,000 queries with answers to share the rich information and benchmarking capabilities it enables. Our first goal is to inspire the research community to try and solve reading comprehension by building great question answering and related models with the ability to carry out complex reasoning. We also aim to gather feedback and learn from the community towards completing the one-million query dataset in the near future.

This dataset has specific value-added features that distinguish itself from previous datasets freely available to researchers. The following factors describe the uniqueness of the MS MARCO dataset:

- All questions are *real, anonymized user queries* issued to the Bing search engine.
- The context passages, which answers are derived from, are extracted from *real Web documents* in the Bing Index.
- All of the answers to the queries are *human generated*.
- A subset of these queries has *multiple answers*.
- All queries are tagged with *segment* information.

The next sections outline the structure, building process and distribution of the MS MARCO dataset along with metrics needed to benchmark answer or passage synthesis and our initial experimentation results.

### 3.1 DATASET STRUCTURE AND BUILDING PROCESS

The MS MARCO dataset structure is described in Table 2 below.

Starting with the real-world Bing user queries we filter them down to only those that are asking for a question (1) and the Web index documents mentioned in Table 2 as data sources, we automati-

| Field | Definition |
|---|---|
| Query | Question query real users issued to the Bing search engine. |
| Passages | Top 10 contextual passages extracted from public Web documents to answer the query above. They are presented in ranked order to human judges. |
| Document URLs | URLs for the top documents ranked for the query. These documents are the sources for the contextual passages. |
| Answer(s) | Synthesized answers from human judges for the query, automatically extracted passages and their corresponding public Web documents. |
| Segment | QA classification tag. E.g., tallest mountain in south america belongs to the ENTITY segment because the answer is an entity (Aconcagua). |

Table 2: MS MARCO Dataset Composition

cally extracted context passages from those documents (2). Then, human judges selected relevant passages that helped them write natural language answers to each query in a concise way (3). Following detailed guidelines, judges used a Web-based user interface (UI) to complete this task (3 and 4). A simplified example of such a UI is shown in figure 2.

A feedback cycle and auditing process evaluated dataset quality regularly to ensure answers were accurate and followed the guidelines. In the back-end, we tagged queries with segment classification labels (5) to understand the resulting distribution and the type of data analysis, measurement and experiments this dataset would enable for researchers. Segment tags include

- NUMERIC
- ENTITY
- LOCATION
- PERSON
- DESCRIPTION (Phrase)

It is important to note that the question queries above are not artificially handcrafted questions based on Web documents but real user queries issued to Bing over the years. Humans are not always clear, concise or to the point when asking questions to a search engine. An example of a real question query issued to Bing is {*in what type of circulation does the oxygenated blood flow between the heart and the cells of the body?*}. Unlike previously available datasets, we believe these questions better represent actual human information seeking needs and are more complex to answer compared to artificially generated questions based on a set of documents.

To solve for these types of questions we need a system with human level reading comprehension and reasoning abilities. E.g., given a query such as {*will I qualify for osap if i'm new in canada*} as shown in figure 2 one of the relevant passages include:

*You must be a 1. Canadian citizen, 2. Permanent Resident or 3. Protected person*

A RC model needs to parse and understand that being new to a country is usually the opposite of citizen, permanent resident, etc. This is not a simple task to do in a general way. As part of our dataset quality control process, we noticed that even human judges had a hard time reaching this type of conclusions, especially for content belonging to areas they were not familiar with.

The MS MARCO dataset that we are publishing consists of four major components:

- *Queries*:These are a subset of user queries issued to a commercial search engine wherein the user is looking for a specific answer. This is in contrast to navigational intent which is another major chunk of user queries where the intent is to visit a destination website. The queries were selected through a classifier which was trained towards answer seeking intent of the query based on human labeled data. The query set was further pruned to only contain queries for which the human judges were able to generate an answer based on the passages that were provided to the judges.

- *Passages*: For each query, we also present a set of approximately 10 passages which might *potentially* have the answer to the query. These passages are extracted from relevant web-pages. The passages were selected through a separate IR (information retrieval) based machine learned system.

- *Answers*: For each query, the data set also contain one or multiple answers that were generated by human judges. The judge task involved looking at the passages and synthesizing an answer using the content of the passages that best answers the given query.

- *Query type*: For each query, the dataset also contains the query intent type across five different categories – (a) description, (b) numeric, (c) entity, (d) person and (e) location. For example, "xbox one release date" will be labeled as *numeric* while "how to cook a turkey" will be of type *description*.This classification is done using a machine learned classifier using human labeled training data. The features of the classifier included uni-gram/bigram features, brown clustering features, LDA cluster features, dependency parser features, amongst others. The classifier was a multi-class SVM classifier with an accuracy of $90.31\%$ over test data.

Since the query set is coming from real user queries, not all queries explicitly contain "what", "where", "how" kind of keywords even though the intents are similar. For example, users could type in a query like "what is the age of barack obama" as "barack obama age". Table 3.1 lists the percentage of queries that explicitly contain the words "what", "where", etc.

| Query contains | Percentage of queries |
|---|---|
| what | 37.7% |
| how | 15.2% |
| where | 4.8% |
| when | 2.2% |
| who | 1.9% |
| why | 1.4% |
| which | 1.4% |

Table 3: Percentage of queries containing question keywords

The following table shows the distribution of queries across different answer types as described earlier in this section.

| Answer type | Percentage of queries |
|---|---|
| Description | 49.3% |
| Numeric | 31.2% |
| Entity | 10.1% |
| Location | 6.3% |
| Person | 3.1% |

Table 4: Distribution of queries based on answer-type classifier

## 4 EXPERIMENTAL RESULTS

In this section, we present our results over a range of experiments designed to showcase characteristics of MS MARCO dataset. As we discussed in section 3, human judgments are being accumulated in order to grow the dataset to the expected scale. Along the time line various snapshots of the dataset were taken and used in thoughtfully designed experiments for validation and insights. With dataset developing, the finalized experiment results may differ on the complete dataset, however, we expect observations and conclusions to be reasonably representative.

We group the queries in MS MARCO dataset into various categories based on their answer types, as described in subsection 3.1. The complexity of the answers varies greatly from category to category. For example, the answers to Yes/No questions are simply binary. The answers to entity questions can be a single entity name or phrase, such as the answer "Rome" for query "What is the capital of Italy".

However, for other categories such as description queries, a longer textual answer is often required to answer to full extent, such as query "What is the agenda for Hollande's state visit to Washington?". These long textual answers may need to be derived through reasoning across multiple pieces of text. Since we impose no restrictions on the vocabulary used, different human editors often compose for the same query multiple reference answers with different expressions.

Therefore, in our experiments different evaluation metrics are used for different categories many of them presented in an earlier paper Mitra et al.. As shown in subsection 4.1 and 4.2, we use accuracy and precision-recall to measure the quality of the numeric answers, and apply metrics like ROUGE-L Lin (2004) and phrasing-aware evaluation framework Mitra et al. for long textual answers. The phrasing-aware evaluation framework aims to deal with the diversity of natural language in evaluating long textual answers. The evaluation requires a large number of reference answers per question that are each curated by a different human editor, thus providing a natural way to estimate how diversely a group of individuals may phrase the answer to the same question. A family of pairwise similarity based metrics can used to incorporate consensus between different reference answers for evaluation. These metrics are simple modifications to metrics like BLEU Papineni et al. (2002) and METEOR Banerjee & Lavie (2005), and are shown to achieve better correlation with human judgments. Accordingly as part of our experiments, a subset of MS MARCO where each query has multiple answers was used to evaluate model performance with both BLEU and pa-BLEU as metrics.

## 4.1 GENERATIVE MODEL EXPERIMENTS

Recurrent Neural Networks (RNNs) are capable of predicting future elements from sequence prior. It is often used as a generative language model for various NLP tasks, such as machine translation Bahdanau et al. (2014), query answering Hermann et al. (2015), etc. In this QA experiment setup, we mainly target training and evaluation of such generative models which predict the human-generated answers given queries and/or contextual passages as model input.

- *Sequence-to-Sequence (Seq2Seq) Model*: Seq2Seq Sutskever et al. (2014) model is one of the most commonly used RNN models. We trained a vanilla Seq2Seq model similar to the one described in Sutskever et al. (2014) with query as source sequence and answer as target sequence.

- *Memory Networks Model*: End-to-End Memory Networks Sukhbaatar et al. (2015) was proposed for and has shown good performance in QA task for its ability of learning memory representation of contextual information. We adapted this model for generation by using summed memory representation as the initial state of a RNN decoder.

- *Discriminative Model*: For comparison we also trained a discriminative model to rank provided passages as a baseline. This is a variant of Huang et al. (2013) where we use LSTM Hochreiter & Schmidhuber (1997) in place of Multilayer Perceptron (MLP).

| | Description | ROUGE-L |
|---|---|---|
| Best Passage | Best ROUGE-L of any passage | 0.351 |
| Passage Ranking | A DSSM-alike passage ranking model | 0.177 |
| Sequence to Sequence | Vanilla seq2seq model predicting answers from questions | 0.089 |
| Memory Network | Seq2seq model with MemNN for passages | 0.119 |

Table 5: ROUGE-L of Different QA Models Tested against a Subset of MS MARCO

| | BLEU | pa-BLEU |
|---|---|---|
| Best Passage | 0.359 | 0.453 |
| Memory Network | 0.340 | 0.341 |

Table 6: BLEU and pa-BLEU on a Multi-Answer Subset of MS MARCO

Table 5 shows the result quality from these models using ROUGE-L metric. While passages provided in MS MARCO generally contains useful information for given queries, the answer generation

nature of the problem makes it relatively challenging for simple generative models to achieve great results. Model advancement from Seq2Seq to Memory Networks are captured by MS MARCO on ROUGE-L.

Additionally we evaluated Memory Networks model on an MS MARCO subset where queries have multiple answers. Table 6 shows answers quality of the model measured by BLEU and its pairwise variant pa-BLEU.

## 4.2 CLOZE-STYLE MODEL EXPERIMENTS

Cloze-style test is a representative and fundamental problem in machine reading comprehension. In this test, a model attempts to predict missing symbols in a partially given text sequence by reading context texts that potentially have helpful information. CNN and Daily Mail dataset is one of the most commonly used cloze-style QA dataset. Sizable progress has been made recently from various model proposals in participating cloze-style test competition on these datasets. In this section, we present the performance of two machine reading comprehension models using both CNN test dataset and a MS MARCO subset. The subset is filtered to numeric answer type category, to which cloze-style test is applicable.

- *Attention Sum Reader (AS Reader)*: AS Reader Kadlec et al. (2016) is a simple model that uses attention to directly pick the answer from the context.

- *ReasoNet*: ReasoNet Shen et al. (2016) also relies on attention, but is also a dynamic multi-turn model that attempts to exploit and reason over the relation among queries, contexts and answers.

|  | Accuracy | |
|  | MS MARCO | CNN (test) |
| --- | --- | --- |
| AS Reader | 55.0 | 69.5 |
| ReasoNet | 58.9 | 74.7 |

Table 7: Accuracy of MRC Models on Numeric Segment of MS MARCO

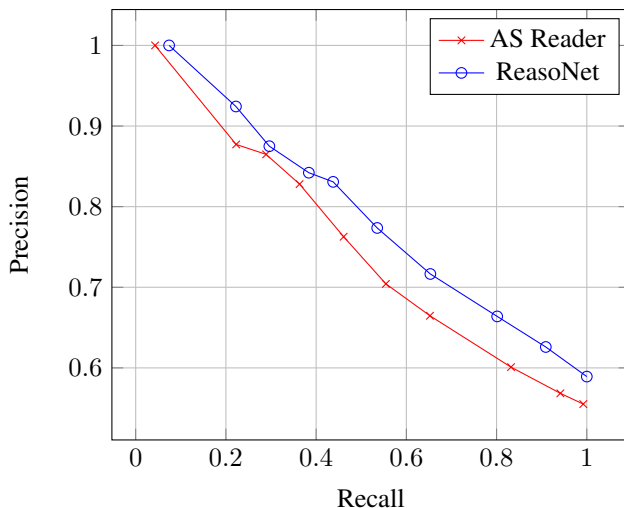

Figure 1: Precision-Recall of Machine Reading Comprehension Models on MS MARCO Subset of Numeric Category

We show model accuracy numbers on both datasets in table 7, and precision-recall curves on MS MARCO subset in figure 1.

## 5 SUMMARY AND FUTURE WORK

The MS MARCO dataset described in this paper above provides training data with question-answer pairs, where only a single answer text is provided via crowdsourcing. This simplicity makes the evaluation relatively easy. However, in the real world, multiple and equally valid answers are possible to a single question. This is akin to machine translation where multiple ways of translation are equally valid. Our immediate future work is to enrich the test set of the current dataset by providing multiple answers. We plan to add 1000 to 5000 such multiple answers in the dateset described in this paper.

Subsequent evaluation experiments on comparing single vs. multiple answers will be conducted to understand whether the model we have built has better resolution with multiple answers. The evaluation metric can be the same METEOR as described in the experiments reported earlier in this paper.

While MS MARCO has overcome a set of undesirable characteristics of the existing RC and QA datasets, notably the requirement that the answers to questions have to be restricted to an entity or a span from the existing reading text. Our longer-term goal is to be able to develop more advanced datasets to assess and facilitate research towards real, human-like reading comprehension. Currently, much of the successes of deep learning has been demonstrated in classification tasks Deng & Yu (2014). Extending this success, the more complex reasoning process in many current deep-learning-based RC and QA methods has relied on multiple stages of memory networks with attention mechanisms and with close supervision information for classification. These artificial memory elements are far away from the human memory mechanism, and they derive their power mainly from the labeled data (single or multiple answers as labels) which guides the learning of network weights using a largely supervised learning paradigm. This is completely different from how human does reasoning. If we ask the current connectionist reasoning models trained on question-answer pairs to do another task such as recommendation or translation that are away from the intended classification task (i.e. answering questions expressed in a pre-fixed vocabulary), they will completely fail. Human cognitive reasoning would not fail in such cases. While recent work is moving towards this important direction Graves et al. (2016), how to develop new deep learning methods towards human-like natural language understanding and reasoning, and how to design more advanced datasets to evaluate and facilitate this research is our longer-term goal.

Figure 2: Simplified passage selection and answer summarization UI for human judges.

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
