# Peer review of "MS MARCO: A Human-Generated MAchine Reading COmprehension Dataset"

_ICLR 2017 — rejected_

[Official Review · AnonReviewer1 · rating 6 · confidence 3 · 16 Dec 2016]
**Need human performance, comparison of distribution of questions / answer-types, comment on automatic metrics**

Summary: The paper proposes a large-scale dataset for reading comprehension, with the final goal of releasing 1 million questions and answers. The authors have currently released 100,000 queries and their answers. The dataset differs from existing reading comprehension datasets mainly w.r.t queries being sampled from user queries rather than being generated by crowd-workers and answers being generated by crowd-workers rather than being spans of text from the provided passage. The paper presents some analysis of the dataset such as distribution of answer types. The paper also presents the results of some generative and some cloze-style models on the MS MARCO dataset.

Strengths:

1. The paper provides useful insights about the limitations of the existing reading comprehension datasets – questions asked by crowd-workers have different distribution compared to that of questions asked by actual users of intelligent agents, answers being restricted to span from the reading text rather than requiring reasoning across multiple pieces of text/passages.

2. MS MARCO dataset has novel useful characteristics compared to existing reading comprehension datasets – questions are sampled from user queries, answers are generated by humans.

3. The experimental evaluation of the existing baseline models on the MS MARCO dataset is satisfactory.

Weaknesses/Suggestions:

1. The paper does not report human performance on the dataset. Human performance should be reported to estimate the difficulty of the dataset. The degree of inter-human agreement will also reflect how well the metric (being used to compute inter-human agreement and accuracies of the baseline models) can deal with variance in the sentence structure with similar semantics.

2. I would like to see the comparison between the answer type distribution in the MS MARCO dataset and that in existing reading comprehension datasets such as SQuAD. This would ground the claim made in the paper the distributions of questions asked by crowd-workers is different from that of user queries.

3. The paper uses automatic metrics such as ROUGE, BLEU for evaluating natural language answers. However, it is known that such metrics poorly correlate with human judgement for tasks such as image caption evaluation (Chen et al., Microsoft COCO Captions: Data Collection and Evaluation Server, CoRR abs/1504.00325 (2015)). So, I wonder how authors justify using such metrics for evaluating open-ended natural language answers.

4. The paper mentions that a classifier was used to filter answer seeking queries from all Bing queries. It would be good to mention the accuracy of this classifier. This will provide insights into what percentage of the MS MARCO questions are answer seeking queries. Similarly, what is the accuracy of the information retrieval based system used to retrieve passages for filtered queries?

5. Please include the description of the best passage baseline in the paper.
  
6. Fix opening quotes, i.e. ” -> “ (for instance, on page 5, ”what” -> “what”).

Review Summary: The paper is well motivated, the use of user queries and human generated answers makes the dataset different from existing datasets. However, I would like to see the human performance on the dataset and quantitative comparison between the distribution of questions obtained from user queries and that of crowd-sourced questions. I would also like the authors to comment on the use of automatic metrics (such as ROUGE, BLEU) in the light of the fact that such metrics do not correlate well with human judgements for tasks such as image caption evaluation.

[Reviewer Comment · AnonReviewer2 · rating 6 · 18 Dec 2016]
**No Title**

This is a dataset paper that brings unique values over existing reading comprehension challenges. Unlike others, MS MARCO is derived from query logs, thus represents real questions that people ask, rather than solicited questions that might be rather artificial in practical settings.

There are potential downsides of using query logs however. It may be that people adapt their language and questions for search engines such that users ask questions that they know current search engines can reasonably answer. Thus, it may be that people limit the complexity of questions or language or both. I think authors could have addressed this concern by being more selective about the query logs, by down-sampling on simple questions that can be easily answered by keyword matching without any sophisticated reading comprehension, and up-sampling more complex questions that require at least paraphrasing and ideally synthesis of information taken from more than one sentences.

It’s great that there are several new efforts to construct large-scale reading comprehension challenges, but my main concern is whether the majority of the questions can be answered through relatively easy text matching without intelligent reading or reasoning.

Also, the paper reads like the authors were running out of time before the deadline. I would appreciate more analytic and quantitative comparisons against other existing datasets, and more insights on the degree of challenges required to handle QAs in MS MARCO. For example, the authors could collect statistics on QAs: (1) exact match exists in the text snippet, (2) paraphrasing is required but otherwise the relevant answer is directly available in the text snippet, (3) requires synthesizing information taken from more than one sentences, (4) requires external knowledge. The author response mentions that (4) is unlikely, but a more formal and complete analysis would be helpful.

[Official Review · AnonReviewer3 · rating 6 · confidence 3 · 21 Dec 2016]

Paper Summary: 
This paper presents a new large scale machine reading comprehension dataset called MS MARCO. It is different from existing datasets in that the questions are real user queries, the context passages are real web documents, and free form answers are generated by humans instead of spans in the context. The paper also includes some analysis of the dataset and performance of QA models on the dataset.

Paper Strengths: 
-- The questions in the dataset are real queries from users instead of humans writing questions given some context.
-- Context passages are extracted from real web documents which are used by search engines to find answers to the given query.
-- Answers are generated by humans instead of being spans in context.
-- It is large scale dataset, with an aim of 1 million queries. Current release includes 100,000 queries.

Paper Weaknesses: 
-- The authors say, "We have found that the distribution of actual questions users ask intelligent agents can be very different from those conceived from crowdsourcing them from the text.", but the statement is not backed up with any study.
-- The paper doesn't clearly present what additional information can today's QA models learn from MS MARCO which they can't from existing datasets. 
-- The paper should talk about what challenges are involved in obtaining a good performance on this dataset.
-- What are the human performances as compared to the models presented in the paper?
-- In section 4.1, what are the train/test splits? The results are for the subset of MS MARCO where every query has multiple answers. How big is that subset?
-- What is DSSM mentioned in row 2, Table 5?
-- The authors should include in the paper how experiments in section 4.2 prove that MS MARCO is a better dataset.
-- In Table 6, the performance of Memory Networks is already close to Best Passage. Does that mean there is not enough room for improvement there?
-- The paper seems to be written in hurry, with partial analysis, evaluation and various mistakes in the text.

Preliminary Evaluation: 
The proposed dataset MS MARCO is unique from existing datasets as it is a good representative of the QA task encountered by search engines. I think it can be a very useful dataset for the community to benefit from. Given the huge potential in the dataset, this paper lacks the analysis and evaluation needed to present the dataset's worth. I think it can benefit a lot with a more comprehensive analysis of the dataset.

[Public Comment · (anonymous) · 04 Jan 2017]
**More details needed for reproducibility**

While this dataset would undoubtedly benefit the community, the paper itself lacks sufficient details to to establish reproducible baselines, which is a key part of a dataset submission in my opinion.

Firstly, the model descriptions are lacking:

1. What exactly is the "best passage" model?
2. What is "a DSSM-alike passage ranking model"?
3. For each model, how does the model handle each case where there are multiple selected passages, a single selected passage, and zero selected passage?

Secondly, the evaluation set on which the metrics are computed are not clearly defined.

1. The authors refer to "a subset of MS MARCO" in all evaluation tables (5, 6, and 7). What examples do these subsets contain? Without this information non of the evaluations are reproducible.
2. How does the author evaluate the case where there are no answers or there are several answers? It is ambiguous how one computes the BLEU or ROUGE scores in these cases.

Lastly, there are no human baselines on this dataset, making the upper bounds unclear (unlike some of the other datasets released).

[Final Decision · Program Chairs · 06 Feb 2017]
**ICLR committee final decision**

Though the dataset is likely to have a large impact on the community, there is a general consensus among the reviewers that the authors could have done a better job characterizing the dataset (lack discussion on answer types, careful comparison with previous datasets, human performance on the dataset). So sadly, though the dataset is potentially very important, the paper does not quite cut it.
 
 Positive:
 -- an important dataset
 -- a good job with establishing baselines
 
 Negative
 -- analysis and discussions are very limited